# Low Dielectric Loss and Multiferroic Properties in Ferroelectric/Mutiferroic/Ferroelectric Sandwich Structured Thin Films

**Zhi-Yong Wu [1],\* and Cai-Bin Ma [2]**

[1]   College of Traditional Chinese Medicine, Southern Medical University, Guangzhou 510515, China
[2]   School of Physics & Optoelectric Engineering, Guangdong University of Technology, Guangzhou Higher Education Mega Centre, Guangzhou 510006, China
\*   Correspondence: wuzhiyong608@163.com; Tel./Fax: +86-20-6278-7160

**Abstract:**   Bismuth ferrite ($BiFeO_3$) has proven to be promising for a wide variety of microelectric and magnetoelectric devices applications. In this work, a dense $(Ba_{0.65}Sr_{0.35})TiO_3(BST)/$ $(Bi_{0.875}Nd_{0.125})FeO_3(BNF)/BST$ trilayered thin film grown on Pt-coated Si (100) substrates was developed by the rf-sputtering. For comparison, single-layered BNF and BST were also prepared on the same substrates, respectively. The results show that the dielectric loses suppression in BST/BNF/BST trilayered thin films at room temperature but has enhanced ferromagnetic and ferroelectric properties. The remnant polarization ($P_r$) and coercive electronic field ($E_c$) were 5.51 $\mu C/cm^2$ and 18.3 kV/cm, and the remnant magnetization ($M_r$) and coercive magnetic field ($H_c$) were 10.1 $emu/cm^3$ and 351 Oe, respectively, for the trilayered film. We considered that the bismuth's volatilization was limited by BST bottom layers making the Bi/Fe in good station, and the action of BST layer in the charge transfer between BNF thin film and electrode led to the quite low leakage current and enhanced multiferroic property. The origin of the mechanism of the highly enhanced dielectric constant and decreased loss tan$\delta$ was discussed.

**Keywords:** multiferroic thin films; low dielectric loss; leakage current; ferromagnetic; ferroelectric

## 1. Introduction

Multiferroic (MF) materials, such as bismuth ferrite $BiFeO_3$ (BFO), have been widely studied due to their potential applications for ferroelectric (FE) memories, piezoelectric sensor, and magnetoelectric devices [1–3]. Due to a large spontaneous polarization, a high Curie temperature ($T_C$ = 1103 K), and a high Neel temperature ($T_N$ = 643 K), BFO has fascinated researchers since it was first synthesized. However, BFO has remained unsuitable, due to its high leakage current and large dielectric loss [4–6]. Magnetic properties can be adjusted by using $Nd^{3+}$ ions instead of larger $Bi^{3+}$ ions in the BFO composition [7], and in order to improve leakage current, Ni, Ti, Sm, Mn, Zn, and Ho doped in BFO thin films [8–14], to improve ferroelectric properties and ferroelectric properties, but there was limited action in reducing leakage current. Due to its nonlinear dielectric properties, $(Ba,Sr)TiO_3$ has been widely used in electronic devices [15–17]; however, there is low remnant polarization and coercive electric field in the $(Ba,Sr)TiO_3$ thin film. In order to improve the electrical properties of MF films, the electrical properties and MF properties of MF films can be controlled by interfacial layer. Miao et al. explained the enhanced fatigue properties in MF $(Ba,Sr)TiO_3/(Bi,La)FeO_3$ thin films [18]; Feng et al. reported the electrical properties of $BiFeO_3$ thin film with a $BaTiO_3$ (or $SrTiO_3$) buffer layer [19]; and Zhang et al. reduced the leakage current density in $Bi_2NiMnO_6$ thin films with double $SrTiO_3$ layers [20]. Guo et al. improved the switching speed of $BiFeO_3$ capacitors by electrodes' conductivity and magnetic

fields [21,22]. The electric field control of magnetism requires deterministic control of the magnetic order, and the magnetoelectric coupling in multiferrous $BiFeO_3$ can be understood [23]. In addition, the others FE/MF heterostructured thin films and FE/MF(FE)/FE trilayered structural thin films were prepared [24–27], but these films still have a high leakage current density and a dissipation factor.

In this work, $Ba_{0.65}Sr_{0.35}TiO_3$, $Bi_{0.875}Nd_{0.125}FeO_3$ and $Ba_{0.65}Sr_{0.35}TiO_3$–$Bi_{0.875}Nd_{0.125}FeO_3$–$Ba_{0.65}Sr_{0.35}TiO_3$ thin films grown on Pt/Ti/SiO$_2$/Si(100) substrates were prepared by radio frequency magnetron sputtering. The ferroelectric, ferromagnetic, and electrical properties of these film samples were investigated. The low dielectric loss, low leakage current density, and multiferroic properties enhancement mechanism of the sandwich structural thin films are also discussed.

## 2. Materials and Methods

The single-layered $Ba_{0.65}Sr_{0.35}TiO_3$ (BST), $Bi_{0.875}Nd_{0.125}FeO_3$ (BNF), and trilayered BST(50 nm)/BNF(50 nm)/BST(50 nm) thin films grown on Pt(111)/Ti/SiO$_2$/Si(100) substrates using radio frequency magnetron sputtering with ceramic targets of BNF and BST [28,29]. When BST and BNF thin films sputter, the maintenance of 500 °C is the ideal substrate temperature, and Ar/O$_2$ (pressure ratio 9:1) and nitrogen are the ideal deposition atmosphere, respectively [30]. We annealed the samples at 650 °C by use of a rapid thermal annealing (RTA) furnace. The preparation of a three-layer heterostructure BST/BNF/BST film was completed. By comparison, a BST and BNF thin film with a thickness of 150 nm was also achieved. For electrical measurements, we adopted a vacuum evaporation to deposited 0.2 mm diameter top gold (Au) electrodes onto the BST and BNF and BST/BNF/BST thin films, respectively. The thickness of the thin films on Pt(111)/Ti/SiO$_2$/Si(100) substrates was measured by a surface profiler (KLA-Tencor P-10, Class One Equipment Inc., Decatur, GA, USA), which was about 150 nm. The crystal structure of the BNF, BST and BST/BNF/BST thin films were characterized by X-ray diffraction (XRD) (D-MAX 2200, Rigaku, Tokyo, Japan) with automated powder diffractometer 1710 (PANalytical B.V., Almelo, Netherlands) in θ–2θ configuration with Cu Kα (λ = 1.5406 Å) radiation. The surface and cross-sectional images were probed by field emission scanning electron microscope (FESEM, S-4800, Hitachi, Tokyo, Japan) with 1 and 5 kV, respectively. We measured the dielectric properties of Au/BST/Pt, Au/BNF/Pt, and Au/BST/BNF/BST/Pt capacitors by means of Agilent 4284A impedance analyzers (Shenzhen Shunyuanda Technology Co., Ltd., Shenzhen, China) under room temperature. FE properties were measured using Precision premier II ferroelectric tester (Agilent Technologies Inc., Santa Clara, CA, USA), and the ferromagnetic properties were characterized by Magnetic Property Measurement system MPMS XL-7 (Quantum Design, San Diego, CA, USA).

## 3. Results and Discussion

Figure 1 illustrates the XRD patterns of the three BNFs, BSTs, and BST/BNF/BST thin films were grown on Pt(111)/Ti/SiO$_2$/Si(100) substrates, respectively. When the BNF film was annealed at 650 °C with nitrogen atmosphere, the film sample (see Figure 1a) crystallized a pure rhombohedral (R3c) distorted perovskite structure [29]. In addition, the BST film was annealed at 650 °C, the film crystallized a pure cubic perovskite structure (see Figure 1b). From Figure 1b,c, it is can be seen that BST was used as buffer-layer, the BNF films grown on Pt/Ti/SiO$_2$/Si(100) substrate with pseudocubic phase was annealed at 650 °C in oxygen atmosphere. From Figure 1, these in-plane lattice parameters of bottom BST, BNF, and top BST thin films were obtained; the results were 4.001(4), 3.961(1), and 3.996(4) Å, respectively for bottom BST, BNF and top BST thin film. Stryukov et al. reported that when the $Bi_{0.98}Nd_{0.02}FeO_3$ thin films growth on the $Ba_{0.8}Sr_{0.2}TiO_3$ buffer-layer with tetragonal phase, the heterostructures $Bi_{0.98}Nd_{0.02}FeO_3/Ba_{0.8}Sr_{0.2}TiO_3$ on (100)MgO substrate has a monoclinic structure with the space group Cc at room temperature [29]. Thus, the BST layer is conducive to the growth of BNF crystals. The BST and BST/BNF/BST thin films display polycrystalline structure and without any secondary phases. Typically, the BNF film is annealed in an oxygen atmosphere and a second phase appears [30]. The crystal structure of the BNF thin films can be controlled by using BST buffer-layer and annealing atmosphere.

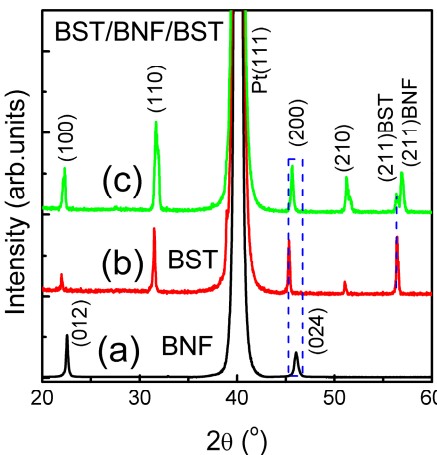

**Figure 1.** XRD patterns of (**a**) BNF, (**b**) BST, and (**c**) BST/BNF/BST thin films on Pt(111)/Ti/SiO$_2$/Si(100) substrates. When BST and BNF thin films sputter, the maintenance of 500 °C is the ideal substrate temperature, and Ar/O$_2$ (pressure ratio 9:1) and nitrogen are the ideal deposition atmosphere, respectively. We annealed the samples at 650 °C by use of a rapid thermal annealing (RTA) furnace.

The FESEM morphologies of BST/BNF/BST heterostructure film on PT/Ti/SiO$_2$/Si(100) substrate annealed at 650 °C for 30 min in oxygen atmosphere was shown in Figure 2. The results show that the structure of the film is dense and uniform. 20 nm is about the average particle size. The thickness of BST at the bottom, BNF at the middle, and BST at the top are ~50, 50, and 50 nm, respectively. The result is consistent with the measurement by the surface profiler.

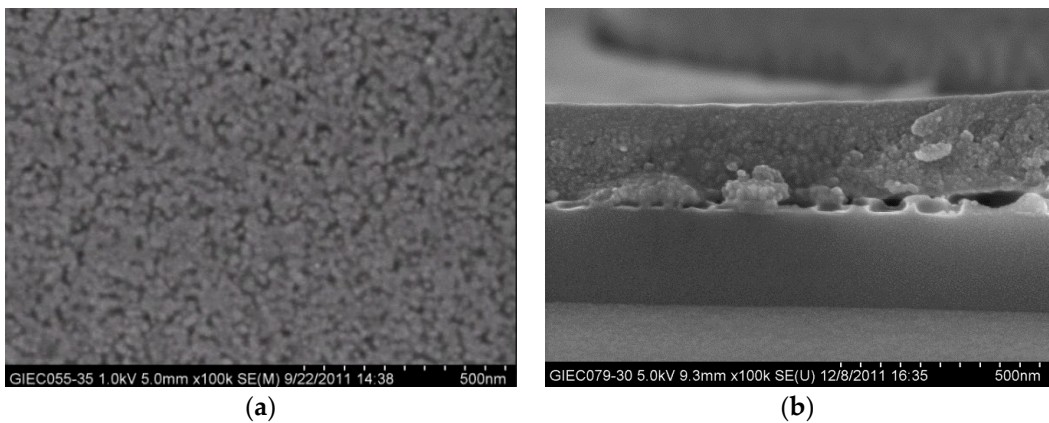

**Figure 2.** The SEM images of surface (**a**) and cross-section (**b**) of the BST/BNF/BST films on Pt(111)/Ti/SiO$_2$/Si(100) substrates annealed 650 °C for 30 min in an oxygen atmosphere.

The dielectric constant ($\varepsilon_r$) and dissipation factor (loss tan$\delta$) were as a function of frequency in the range of 100 Hz–2 MHz for the three thin film capacitors of the Au/BNF/Pt, Au/BST/Pt, and Au/BST/BNF/BST/Pt at RT. Figure 3 shows the dielectric properties of those thin film capacitors. For the Au/BNF/Pt film capacitor, the $\varepsilon_r$ and loss tan$\delta$ decrease from 68 to 59 and from 0.141 to 0.052 with increasing frequency from 100 Hz to 2 MHz, respectively. In addition, for the Au/BST/Pt film capacitor, the $\varepsilon_r$ decrease and loss tan$\delta$ increase from 341 to 331 and from 0.017 to 0.018 with increasing frequency from 100 Hz to 2 MHz, respectively. However, for the sandwich structure Au/BST/BNF/BST/Pt film capacitor, the $\varepsilon_r$ and loss tan$\delta$ slightly decrease from 277 to 275 and 0.0045 to 0.0032 with increasing frequency from 100 Hz to 2 MHz. At 100 kHz, the $\varepsilon_r$ and loss tan$\delta$ were 61 and 0.052, 334 and 0.018, and 275 and 0.0033, respectively for the three film capacitors of Au/BNF/Pt, Au/BST/Pt, and Au/BST/BNF/BST/Pt. Use the single and double layers of BST thin films, the dielectric loss is suppressed in the Au/BST/BNF/BST/Pt thin film capacitors, the results are consistent with those of leakage current

measurements. Most of the dielectric loss comes from conductivity loss. The equivalent resistance in multilayer thin films is equal to that in each layer, which reduces the dielectric loss of multilayer thin films [31].

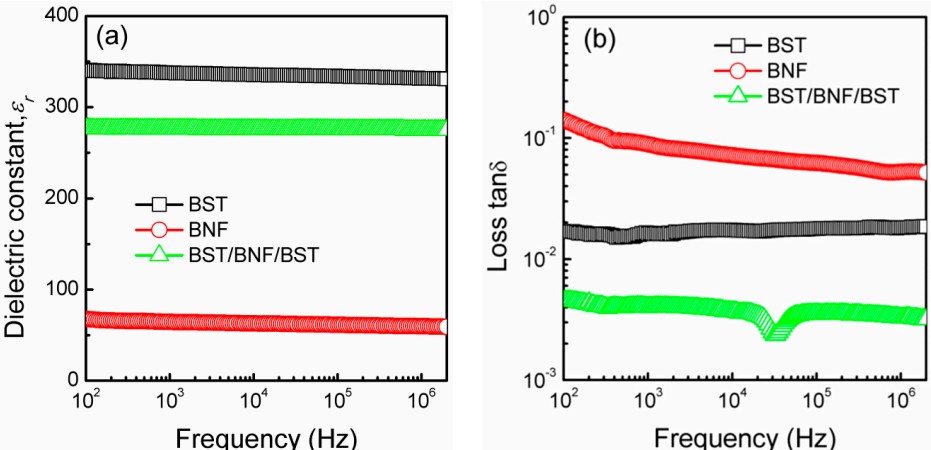

**Figure 3.** The dielectric constant (**a**) and loss tanδ (**b**) versus frequency for BST, BNF and BST/BNF/BST films on Pt(111)/Ti/SiO$_2$/Si(100) substrate measured at room temperatures.

In general, the equivalent capacitor of multilayer structure thin films may be expressed by series capacitors [32–34]. For the $\varepsilon_r$ value of multilayer thin films, the effective capacitor was constructed by three capacitors, include of top $C_{BST}$, $C_{BNF}$, and bottom $C_{BST}$ capacitors, respectively. Compared the reported about dissipation factor in (Pb,Sr)TiO$_3$/(Bi,La)FeO$_3$/(Pb,Sr)TiO$_3$ trilayered structure [35], there is extremely low loss tanδ in the BST/BNF/BST thin films. In Figure 3b, the multilayers thin films have very low loss tanδ, the degree of the loss tanδ was ~10$^{-3}$ from 100 Hz to 2 MHz. The bismuth's volatilization was limited in annealed processing by bottom and top BST layers, making the Bi/Fe in good station in BNF layer; it also improved the technology of preparation bringing down the defect rate of BST/BNF/BST film, and reducing the number of free-carriers.

In Figure 4, the typical polarization (*P*)–electric field (*E*) hysteresis loops for the Au/BNF/Pt and Au/BST/BNF/BST/Pt thin film capacitors at RT and 250 Hz. The inset of Figure 4 shows an unsaturated loose *P*–*E* loop of the Au/BST/Pt thin layer capacitor, the remnant polarization ($P_r$), saturation polarization ($P_s$) and coercive field ($E_c$) were 0.223 μC/cm$^2$, 1.61 μC/cm$^2$, and 20.3 kV/cm, respectively. Bulk (Ba$_{0.65}$Sr$_{0.35}$)TiO$_3$ material is cubic phase at room temperature, but due to the relaxor ferroelectric characteristics of small grain (Ba$_{0.65}$Sr$_{0.35}$)TiO$_3$, it still has ferroelectric properties at room temperature and higher temperatures [36]. From Figures 1 and 4, the BST is cubic phase at room temperature, and the ferroelectric properties comes from the polar nanoregions [37,38]. For the Au/BST/BNF/BST/Pt trilayered film capacitor, the loop shows a slim shape and reaches saturation at low electric field (about 145 kV/cm). It exhibited the $P_r$, $P_s$ and $E_c$ were 3.77μC/cm$^2$, 18.5 μC/cm$^2$, and 21.7 kV/cm, 5.06 μC/cm$^2$, 25.0 μC/cm$^2$, and 18.3 kV/cm, respectively for Au/BNF/Pt and Au/BST/BNF/BST/Pt thin film capacitors. These look like a banana ferroelectric for BNF and BST thin films [39], because the applied electric field is too low. When applied high electric field (>600 kV/cm), the (Bi,Nd)FeO$_3$ thin films show good ferroelectricity [40,41]. From Figure 4, there a gap in the *P*–*E* loop in Au/BNF/Pt thin film capacitor, indicating that the degree of leakage current in the Au/BNF/Pt film capacitor was much higher than that of the Au/BST/BNF/BST/Pt film capacitor. The ferroelectric enhancement in the BST/BNF/BST was owed to the coupling between BST and BFO thin films [18,35]. In addition, the remnant polarization, permittivity, and resistivity of BFO thin films have direct correlation with the Bi/Fe ratio [42]. In this work, the bismuth's volatilization was inhibited by BST layers, which makes the Bi/Fe ratio in preferable situation and brings down the defect rate of the film improved the remnant polarization and saturation polarization.

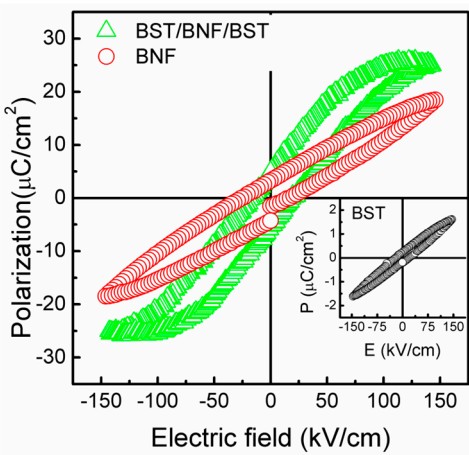

**Figure 4.** The typical *P–E* hysteresis loops for the BNF and BST/BNF/BST films on Pt/(111)Ti/SiO₂/Si(100) substrate at room temperature and 250 Hz, the inset is the *P–E* hysteresis loop of single layer BST thin film.

Figure 5 shows the leakage current density (*J*)–electric field (*E*) characteristics of BST/BNF/BST and BNF film on (111) Pt/Ti/SiO₂/Si substrate. The current demsity (*J*) of BST/BNF/BST is ~$10^{-8}$A/cm² at RT and 100 kV/cm. Which is lower than that of the BNF film. In addition, in the BNF, the *J* value increases rapidly with an increasing *E* value. Scott et al. reported that the *J* value of PZT films can be reduced by decreasing oxygen vacancies (OVs) [43], and they reduced the $Fe^{3+}$ valance state in $BiFeO_3$ film by annealed in nitrogen atmosphere [29,44]. In the quite low leakage current system, films annealed in oxygen can be reduced OVs in BST layers and the BST layers cut off the BNF layer with air reducing the $Fe^{3+}$ valance state. Thus, the *J* value was low due to reduced OVs and $Fe^{3+}$ valance state. Moreover, the interface of ferroelectric BST layer and ferromagnetic BNF thin films plays an important role in the charge transfer between thin film and electrode interface and reducing the *J* value.

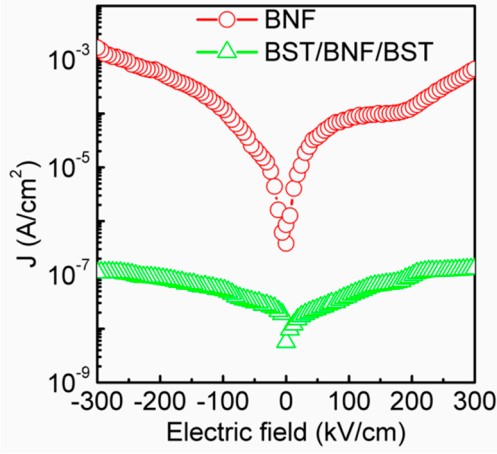

**Figure 5.** *J–E* characteristics of BNF and BST/BNF/BST films on Pt(111)/Ti/SiO₂/Si(100) substrate measured at room temperature.

Figure 6 shows the magnetization (*M*)–magnetic field (*H*) loops of the BNF and BST/BNF/BST films on Pt(111)/Ti/SiO₂/Si(100) substrates at RT. For the single layer BNF film, the inset of Figure 6 shows the loop in a slim shape and reaches saturation at low magnetic field (about 5 kOe). In the BST/BNF/BST thin film, there are higher remnant magnetization ($M_r$) and the lower coercive magnetic field (about 10.1 emu/cm³ and 351 Oe). The value of $M_r$ is slightly larger than tha of the saturated magnetizations ($M_s$ = 7.14 emu/cm³) for the PST/BLF/PST [35]. Thakuria and Joy reported that the $M_s$ and $M_r$ values of the nanocrystalline $Bi_{0.875}Nd_{0.125}FeO_3$ are ~1.3 and 0.20 emu/g, respectively [45]. Ederer et al. reported

that the OVs can alter the magnetization slightly in the heterostructure $BiFeO_3/Ba_{0.25}Sr_{0.75}TiO_3$, and lead to the formation of $Fe^{2+}$ [46]. The BST/BNF/BST film has a more optimized ferromagnetic than single BNF thin film that might be attributed to the bismuth's volatilization when inhibited by BST layers, bringing down the defect rate of BNF film and making the Bi/Fe in good station; the BST layers advanced the superexchange coupling of ferrite or double exchange of $Fe^{2+}$ and $Fe^{3+}$ ions and the incorporation of OVs in the BNF thin film play a key role in enhanced magnetization in BST/BNF/BST thin film.

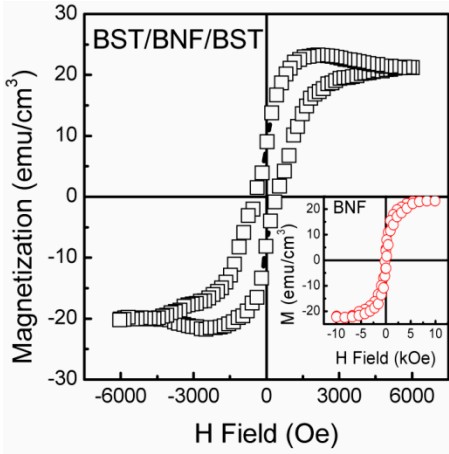

**Figure 6.** The *M–H* hysteresis loop of the BST/BNF/BST film on Pt(111)/Ti/SiO$_2$/Si(100) substrate at room temperature, the inset is the *M–H* loop of single layer BNF thin film.

## 4. Conclusions

The sandwich structured BNF film with double BST layers grown on a Pt(111)/Ti/SiO$_2$/Si(100) substrate was produced by radio frequency magnetron sputtering. The dielectric loss was suppressed and enhanced the multiferroic properties, respectively in the trilayered the films. The bismuth's volatilization by BST layers was limited and brought the Bi/Fe ratio to a desirable situation, maybe leading to quite low leakage current, dielectric loss, and high remnant polarization. It improved the technology of preparation, and by bringing down the defect rate of the BST/BNF/BST film, the leakage current and loss tan δ were ~$10^{-8}$ A/cm$^2$ at 200 kV/cm and ~$10^{-30}$, respectively. Moreover, the low leakage current density may be related to the role of BST in charge transfer between BNF and electrodes, involving the coupling reaction between BST and BNF films.

**Author Contributions:** Conceptualization, Z.-Y.W.; Formal Analysis, Z.-Y.W. and C.-B.M.; Investigation, Z.-Y.W. and C.-B.M.; Writing—Original Draft Preparation, Z.-Y.W. and C.-B.M.; Writing—Review and Editing, Z.-Y.W. and C.-B.M.

**Funding:** This research was funded by the "Guangdong Natural Science Fund Project of China (No. 2015A030313265)".

**Conflicts of Interest:** The authors declare no conflict of interest.

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
