# Peer review of "Low Dielectric Loss and Multiferroic Properties in Ferroelectric/Mutiferroic/Ferroelectric Sandwich Structured Thin Films"

_coatings, doi:10.3390/coatings9080502_

Round 1

Reviewer 1 Report

The authors study the “Low Dielectric Loss an Multiferroic Properties in Ferroelectric/Mutiferroic/Ferroelectric Sandwich Structured Thin Films.” Overall study of the manuscript is very consistent.The compare their study by X-ray diffraction, SEM, Ferroelectric, dielectric impedance and Magnetic. 

However, still I found some concern in the manuscript. 

1.    The introduction is not comprehensive and lacks a clear statement why their outcome is important for scientific community. 

2.     The introduction part did not refer to the important developments multiferroics. I recommend, author should add some references of BFO multiferroics which are studied on the base of its applications (e.g. APL 103, 022905 (2013); APL Mat 2, 096107 (2014); Nature comm. 9, 3764 (2018)). 

Author Response

   The introduction is not comprehensive and lacks a clear statement why their outcome is important for scientific community.

Response:  Its scientific significance is illustrated in the introduction.

    The introduction part did not refer to the important developments multiferroics. I recommend, author should add some references of BFO multiferroics which are studied on the base of its applications (e.g. APL 103, 022905 (2013); APL Mat 2, 096107 (2014); Nature comm. 9, 3764 (2018)).

Response: Thanks you for your comments. We added references. Please see [21-23].

Reviewer 2 Report

The authors report the preparation and characterization of BST/BNF heterostructured films and compare them with single component BST and BNF films. There are places where the manuscript is muddled and where it is hard to pull out the authors conclusions. This is particularly true in the discussion of the XRD results. Overall revisions are necessary before the manuscript can be published.

1.       In places the introduction lacks detail which enables the reader to put the current work in to the context of the current state-of-the-art. For example it states that electrical properties where reported BiFeO3 films with BT or ST buffer layers but no information is provided as to what was actually observed.

2.       The motivation for why these particular films is not clear.

3.       Please add the accelerating voltage to the experimental section for the FESEM.

4.       In the XRD data a peak is labelled as BNF which does not appear in the pure BNF data? Is this correct? Additionally there is also clear splitting of the 210 peak which would suggest that the structure of the BST/BNF/BST film is far more complex than a simple cubic arrangement as suggested. Please add the annealing atmosphere to the figure 1 caption. In fact the general discussion of this data in the manuscript is muddled with the mention of multiple atmospheres and needs to be clearer.

5.       The PE loops presented for BST and BNF do not show any FE character and are typical of lossy dielectrics. (see the work by Sinclair or Ferroelectrics goes Banana’s by Scott). The discussion of these properties needs to be revised. It is interesting that the BST/BNF/BST does show some weak FE character but this is obscured by the BNF data and the use of points it may be better to show as a line.

Author Response

In places the introduction lacks detail which enables the reader to put the current work in to the context of the current state-of-the-art. For example it states that electrical properties where reported BiFeO3 films with BT or ST buffer layers but no information is provided as to what was actually observed.

Response: More details are provided in the introduction.

The motivation for why these particular films is not clear.

Response: The grain size is too small to see the details clearly.

Please add the accelerating voltage to the experimental section for the FESEM.

Response: It has been added to the experiment section.

In the XRD data a peak is labelled as BNF which does not appear in the pure BNF data? Is this correct? Additionally there is also clear splitting of the 210 peak which would suggest that the structure of the BST/BNF/BST film is far more complex than a simple cubic arrangement as suggested. Please add the annealing atmosphere to the figure 1 caption. In fact the general discussion of this data in the manuscript is muddled with the mention of multiple atmospheres and needs to be clearer.

Response: Figure 1. XRD patterns of (a) BNF, (b) BST and (c) BST/BNF/BST thin films on Pt(111)/Ti/SiO2/Si(100) substrates. During the sputtering process, the substrate temperature was kept at 500 C, and the depositing atmosphere is Ar/O2 (pressure ratio is 9:1) and nitrogen, respectively for BST and BNF thin films. The samples were then annealed at 650 C by a RTA furnace under oxygen atmosphere.  

      The PE loops presented for BST and BNF do not show any FE character and are typical of lossy dielectrics. (see the work by Sinclair or Ferroelectrics goes Banana’s by Scott). The discussion of these properties needs to be revised. It is interesting that the BST/BNF/BST does show some weak FE character but this is obscured by the BNF data and the use of points it may be better to show as a line.

Response: It looks like a banana ferroelectrics for BNF and BST thin films [39], because the applied electric field is too low. When applied high electric field (> 600 kV/cm), the (Bi,Nd)FeO3 thin films show good ferroelectricity [40,41]. Figure 4 has been modified. 
